# A Complete Solution for Ultra-Wideband Based Real-Time Positioning

**DOI:** 10.3390/s25154620

**Published:** 2025-07-25

**Authors:** Vlad Ratiu, Ovidiu Ratiu, Olivier Raphael Smeyers, Vasile Teodor Dadarlat, Stefan Vos, Ana Rednic

**Affiliations:** 1Computer Science Department, Technical University of Cluj-Napoca, 400114 Cluj-Napoca, Romania; vasile.dadarlat@cs.utcluj.ro (V.T.D.); ana.rednic@cds.ro (A.R.); 2Control Data Systems SRL, 400267 Cluj-Napoca, Romania; ovidiu.ratiu@cds.ro (O.R.); stefan.vos@cds.ro (S.V.); 3European Space Agency, 2201 AZ Noordwijk, The Netherlands; olivier.smeyers@esa.int

**Keywords:** ultra-wideband, RTLS, TDoA, ToF, positioning, real time, machine learning, AutoGluon

## Abstract

Real-time positioning is a technological field with a multitude of applications, which expand across many scopes: from positioning within a large area to localization within smaller spaces; from locating people to locating equipment; from large-scale industrial or military applications to commercially available solutions. There are at least as many implementations of real-time positioning as there are applications and challenges. Within the domain of Radio Frequency (RF) systems, positioning has been approached from multiple angles. Some of the more common solutions involve using Time of Flight (ToF) and time difference of arrival (TDoA) technologies. Within TDoA-based systems, one common limitation stems from the computational power necessary to run the multi-lateration algorithms at a high enough speed to provide high-frequency refresh rates on the tag positions. The system presented in this study implements a complete hardware and software TDoA-based real-time positioning system, using wireless Ultra-Wideband (UWB) technology. This system demonstrates improvements in the state of the art by addressing the above limitations through the use of a hybrid Machine Learning solution combined with algorithmic fine tuning in order to reduce computational power while achieving the desired positioning accuracy. This study presents the design, implementation, verification and validation of the aforementioned system, as well as an overview of similar solutions.

## 1. Introduction

The scope of this study is to present SatUWB, which is a novel, fully implemented Real-Time Locating System (RTLS) using UWB radio technology both for positioning and communications. This study also presents the state of the art concerning the positioning domain within the RF context and the testing and validation results of the aforementioned system.

The SatUWB system was designed to offer real-time positioning services, both indoors and outdoors. As such, it can extend the capabilities of Global Navigation Satellite System (GNSS)-based positioning strategies by providing real-time locations of tags using known coordinates of anchors. The coordinates of the anchors can either be preprogrammed, in the case of fixed-position anchors, or received from GNSS services, in the case of mobile anchors. As such, the SatUWB system can also function within the domain of indoor positioning by extending the capabilities of GNSS services, which function poorly or not at all in indoor situations.

The authors surveyed the state of the art and concluded that for use cases involving a large number of tags with limited power capabilities and high-frequency positioning refresh rates, the physical communication layer shall be based on UWB, and the architectural solution shall be based on TDoA.

Furthermore, one significant challenge of existing UWB/TDoA-based RTLS has been identified in the area of computational power for high-speed algorithmic processing.

This study shows the improvements introduced by SatUWB over other UWB-based RTLSs by applying Machine Learning as a preprocessing step before applying the positioning algorithms.

Introducing the Machine Learning (ML) techniques as a precursor to algorithmic processing is one of the main contributions of the solution presented and constitutes an evolution compared to the state of the art as described in [1,2,3,4,5,6]. More specifically, the use of ML allows for positioning of the tags within 1 m × 1 m tiles, therefore allowing the positioning algorithms (Gradient Descent or Least Squares) to (a) have a relatively accurate starting point in the center of the respective tile, and (b) greatly reduce the area in which the algorithms are run, therefore reducing the overall complexity of the problem to be solved. While ML has been used in other UWB location systems, as described in [4], these usages are conceptually different than the approach investigated in this study.

Another significant contribution constitutes the specific end-to-end validation and performance assessment for outdoor deployments. While the vast majority of published works address only indoor use cases, the end-to-end functional validation in the current study was performed both in indoor and outdoor conditions, and the results show stable, repeatable performance.

## 2. Materials and Methods

### 2.1. Related Work

In the following section, we present a brief overview of related technologies and indoor UWB-based RTLS approaches.

The case for using UWB is frequently made by referencing the UWB technology’s resistance to interference, resistance to multipath fading, accurate positioning and potentially lower power consumption than other technologies [7,8,9,10].

UWB is an amendment to the 802.15.4 standard [11] that describes an RF wireless communication technique based on very short (1 to 2 ns) impulses. Some of the most important characteristics of this technology, as specified by the UWB hardware selected for implementation, are as follows:Short impulse transmission technique.A span of six RF bands from 3.5 GHz to 6.5 GHz.Support for data rates of 110 kbps, 850 kbps and 6.8 Mbps.Short on-air time due to the high data rates.Low power consumption and extended battery lifetime.2 ns impulse results in 500 MHz channel bandwidth.Ability to deal with severe multipath environments.Ideal for highly reflective RF environments.

UWB has already demonstrated advancements within the domain of indoor positioning systems [1,6,12]. It is within this that we aim to place our solution among the general context of the state of the art and, while acknowledging the related work, provide a foundation for demonstrating our contributions to the domain.

There are multiple complete systems to be found within the positioning domain, each with its own advantages and drawbacks. A notable approach is presented in [2]. The authors describe the implementation of an indoor navigation system used for aiding visually impaired people. Much like the approach described in this study, it uses UWB for positioning. The paper also presents remarkable similarities in the sense that the use case is similar. The paper briefly presents an overview of relevant systems, developed at the time of writing, with similar use cases.

A more comprehensive overview of indoor positioning techniques using UWB is given in [3]. The authors classify positioning algorithms based on the estimating measurements used, namely time of arrival (TOA), angle of arrival (AoA), received signal strength (RSS), time difference of arrival (TDoA) and hybrid algorithms. The authors describe each approach in detail based on the specific algorithms implemented in the state of the art. The paper proceeds to mention one of the most important challenges when using TDoA-based systems, namely that the receivers must cooperate in order to compute the position of the transmitters, which increases the computational bandwidth significantly when compared to other approaches. This is one of the areas where our approach provides novelty to the state of the art. In addition to the low-power aspects of UWB, using TDoA further builds upon that [4]. Multiple approaches were considered when developing the system presented in this study before selecting TDoA as the best possible approach.

When developing an indoor positioning system, a recurrent question, as mentioned in [5], is that of the maximum number of available tags as trackable assets. The study demonstrates that that number is in the hundreds. As will be seen, we manage to implement a system that supports multiple tags, greatly increasing the capabilities towards the end user. As suggested, the system presented in this study uses a TDoA-based implementation.

The authors agree with the general conclusion of the state of the art, which is that the combination of UWB as a physical communication medium and TDoA as a system architectural basis is the optimal solution for low-power, high-scalability RTLS.

One of the main challenges of state-of-the-art TDoA-based RTLS has been identified by the authors as the computational power required to run the various positioning algorithms considered. This challenge is addressed in the current study by means of introducing a Machine Learning preprocessing step before applying the positioning algorithm.

The usage of ML techniques as a preliminary computational step is an original contribution of the authors, which differs from other ML applications described in the literature. For example, the authors of [4] used ML for distinguishing between Line-of-Sight (LoS) and Non-Line-of-Sight (NLoS) situations. This approach has merit but is radically different than how ML is used in the solution presented here.

Another thing that is apparent in the analysis of the subject literature is that published work tends to study the use case of indoor deployments, and there is little information about using UWB for positioning in outdoor installations. The outdoor use case has some particularities when compared with indoors, such as longer distances, in addition to the inherent weatherproof requirements and sometimes challenging communications to remote locations. The solution presented in this study addresses this challenge, and the results for outdoor testing are presented alongside the results for indoor testing.

Other challenges exist as well, such as the accuracy of the clock synchronization between the anchors. However, these are not addressed in the present study and remain a subject for future work. Still, an assessment was performed about the stability obtained for the clock synchronization between anchors, as this is an essential element in TDoA systems. This assessment is presented in Section 2.4.3.

### 2.2. System Architecture

Figure 1 presents a block diagram of the system. The system is composed of multiple UWB tags, which transmit periodic messages. These messages are received by several anchors, which are synchronized in time and relay the received timestamps to a Central Localization Engine (CLE) via a master anchor connected to a gateway. The CLE determines the position of the transmitting tag using a combination of traditional algorithms and ML techniques. In the following sections, each component will be described separately.

### 2.3. UWB Tags

UWB tags are devices that periodically transmit a short message, called a blink message, which contains the tag’s ID for identification and a sequence number to avoid duplication. For this study, the authors used commercial UWB tags manufactured by Control Data Systems (CDS), as shown in Figure 2.

### 2.4. UWB Anchors

There are two types of anchors in the system: master and slave. The master anchor periodically sends synchronization messages to the slave anchors and they, in turn, send tag timestamp information to the master anchor, all in a well-defined manner. In addition, all anchors receive National Marine Electronics Association (NMEA) messages from the GNSS module and record their absolute coordinates every second. For this study, the authors used commercial UWB anchors manufactured by CDS, as shown in Figure 3.

#### 2.4.1. Master Anchor

The master anchor listens to blink messages from tags and, in addition, sends its own messages, used for clock compensation. The anchor functions in reception mode until it needs to send a message. Every 150 milliseconds, the anchor composes a message that contains its transmission-programmed timestamp and sets the radio to send the message at that exact time. After the message is successfully sent, the timer is reset and the whole procedure begins again.

In addition to the role of the clock synchronizer of the network, the master anchor is also responsible for sending all the received tag information from other anchors to the CLE. It does this by using the same interface that is used for communicating the provisioning commands, using the same API format. Upon receiving a tag timestamp message with the parent anchor ID matching its own ID, the master anchor will insert the relevant information in a buffer and notify the Universal Asynchronous Receiver Transmitter (UART) that it has data to send.

#### 2.4.2. Slave Anchor

The slave anchor listens to blink messages from the tags and to synchronization messages from the master anchor or tag timestamp messages. When a message is received from the parent anchor, the slave anchor saves its own reception timestamp of the received message and the transmission timestamp from within the payload. Using these two values, alongside the distance between the anchors (which is known from the GNSS coordinates), the clock skew between the two anchors can be calculated.

#### 2.4.3. Anchor Desynchronization

To estimate the anchor desynchronization value, one tag was placed at known equal distances between two anchors. For each blink emitted by the tag, the two anchors reported their respective timestamps. The difference between the timestamps represents desynchronization. The measurements were taken continuously during a 24 h timeframe, and the results are shown in Figure 4. The average desynchronization time was 258 ps.

### 2.5. Positioning Using ML

The authors of [4] provide an overview of various ML algorithms that may be used in indoor positioning techniques while presenting an evaluation for accuracy, precision, running time and other metrics. The authors conclude that whilst the use of ML for positioning using UWB provides a measurable improvement over other approaches, some of the limitations include the availability of training data, the training time efficiency, extensibility and scalability, variability, energy consumption, and, finally, map construction and route planning.

The project presented in this study builds upon two previous activities [1,13]. Whilst the former presents the results of an indoor positioning system, which uses traditional algorithms, the second prototypes the ML learning approach, which is presented in this study.

Figure 5 illustrates the process of creating and updating a model in successive steps:Data are first split into two datasets for training and testing (steps 1a and 1b);The training dataset is used as input for different algorithms (step 2a and 3a), while the testing dataset is used to validate the output (step 2b and 3b);The model is created and updated continuously (step 3c);The final model is saved (step 5) and can be used for predictions by providing data as input.

Building on the hardware and software of the first approach and merging the second approach within a hybrid positioning algorithm lead to the system presented in the following sections.

### 2.6. Software Application

The application software side of the system consists of multiple components, as shown in Figure 6. These are presented in detail in the following sections.

The software application consists of the following components:Training and Evaluation Engine (TEE): generates simulations to help training of the predictor.Amazon Web Services (AWS): represents the cloud-based training infrastructure, which includes Amazon SageMaker and AutoGluon.AG Predictor: the AutoGluon predictor, which contains the stack of previously trained ML models, including an aggregated model, in the form of an automatically yet transparently generated weighted ensemble.Predictor Container: represents the various software dependencies and management software responsible for deploying and running the model on the CLE, using the AWS server.Preprocessing Module: responsible for any data handling before feeding data to the predictor.Postprocessing module: responsible for any activities related to positioning after the predictor runs (such as Gradient Descent or Least Squares).Data Output/Graphical User Interface (GUI): represents the presentation layer of the whole system.UART: represents the interface on which the master anchor passes data to the CLE.Master Anchor: acquires data from all other anchors, calculates TDoA values and acts as a gateway from the UWB network.

The TEE is commercial software by CDS used to generate the training cases for the AG predictors. It is a Java-based visual application, which takes as input the number and positions of anchors in absolute coordinates, among other data, and uses these to generate .csv files. These .csv files are then manually uploaded to the AWS cloud.

The AWS cloud-based backend system functions as a platform that both stores relevant data (such as the training .csv files) and runs a SageMaker container that allows for the creation of an environment for AG to function.

AutoGluon is an AutoML package, an open-source toolkit developed by Amazon Web Services (AWS). Other AutoML packages include MLBox, Auto-sklearn and Auto-PyTorch. Version 1.3.1 of Autogluon was used in the development of this product.

AutoGluon supports automatic ML on tabular, text and image data. Since the objective of this document involves tabular prediction, the following section briefly describes AutoGluon’s tabular approach, features and general design.

AutoGluon-Tabular is described in [14]. The authors present AutoGluon as contrasting with other AutoML frameworks (at the time of writing) by stacking multiple ML models and ensembling them instead of focusing on model/hyperparameter selection. Furthermore, the authors make a case for AutoGluon by mentioning that previous approaches, through their focus on selecting the best model and performing hyperparameter optimization through brute-force searches, are computationally intensive, as opposed to the stacking approach used by AutoGluon. Another contribution of [14] to the state of the art is an experimental comparative study of six AutoML frameworks from which AutoGluon demonstrates optimizations in accuracy and training time allocation.

AutoGluon data processing is split up into two consecutive phases: a model-agnostic preprocessing stage that transforms all inputs (e.g., data categorization) and a model-specific preprocessing stage that is only applied to a specific copy of the data on a model-by-model basis. Missing discrete variables are still retained, which allows AutoGluon to handle them at test time, but they are added to an additional *Unknown* category.

The authors of [14] mention that AutoGluon possesses an edge over alternative AutoML approaches in that it uses stack ensembles in multiple layers, not unlike a deep learning neural network. Some of the benefits of using AutoGluon include the following:Simplicity: training and evaluation of ML models can be achieved with a few lines of code.Robustness: there is no initial need for heavy-duty data manipulation (although it can be useful for increasing precision).Fault tolerance: intermediate steps during training can be individually inspected and training can be resumed even in interrupted.Customizability: AutoGluon supports custom models. This is achievable by implementing the *_preprocess* and *_fit* methods.Adaptability: AutoGluon works with both Amazon AWS and Microsoft Azure; being open source, a custom AutoGluon configuration is also theoretically possible, as long as the corresponding APIs are respected.

The models currently supported by AutoGluon are featured in Table 1. The selected model used for positioning is described in detail in the following sections.

### 2.7. Model Training

The following section presents different concepts used for model training as well as different methods of evaluating a model, based on different metrics. Building a model using automatic Machine Learning consists of two phases:Training phase: this phase consists of determining optimal functional parameters related to an ML model (e.g., weights and biases).Evaluation phase: this phase consists of comparing ML models based on a specific metric in order to determine the best-performing one.

Some very important terms used when working with an ML model are as follows:Model weights are a set of values embedded in a model used to make predictions.Model training is the process of applying various techniques to find the optimal combination of weights.Features consist of input data (numerical or categorical; categorical features generally need to be normalized since, in their raw form, they are difficult to compare); the values of categorical features are also called categories.Labels/targets represent the output data.

An important phase is data splitting, which occurs before training. As such, data splitting consists of the following steps, as seen in Figure 7:The original set is split into two sets: dataset and training set.The training set is split into two sets: the training set (used for all ML models) and a validation set (used to tune the model).The test set is used to evaluate the performance of the model.

AutoGluon makes use of the weighted ensemble model described in [15]. This approach brings several benefits to the system as a whole. When dealing with a specific problem in which some models may have better performance than others, rather than just adding all models to the ensemble, even those that have poor performance, the authors use a forward stepwise selection approach, in which for a fixed number of iterations, only the model which maximizes the ensemble performance is added to the ensemble; this performance is evaluated by averaging the individual performance with the ensemble performance. Thus, a subset of models is produced, which helps ensure that no particular model lowers the precision of the whole ensemble.

### 2.8. Area Coverage

The maximum distance between anchors varies from roughly 100 m (line of sight, outdoor scenario) down to 20–30 m (multiple obstacles and reflecting areas, indoor scenario). From the perspective of the computational power, which is the focus of this study, the coverage area is divided into equal-sized rectangles, and their number has no impact on the resources needed to run the positioning algorithms and no impact on the Machine Learning preprocessing phase, except for the AI model training, performed at the beginning of the deployment. From the perspective of clock synchronization, the desynchronization between distant, multiple hop anchors and the master anchor will introduce positioning errors, which, at some point, will go over the targeted value. However, this is not the focus of the current study, and the authors intend to research these limitations along with optimization techniques during future work.

The presented weighted ensemble model proposes the use of ten metrics, namely ACC, RMS, MXE, LFT, BEP, FSC, APR, ROC Area, CAL and SAR. The authors mention that so many metrics are taken into consideration since their faring is dependent on learning methods and the situations in which they are used.

The algorithm also normalizes the performances of the various metrics on a [0, 1] scale, so that they can more easily be compared and evaluated.

The authors conclude that when using the weighted ensemble approach with the aforementioned ten performance metrics, on seven test scenarios, the ensemble algorithm outperformed all single models involved (KNN, ANN, DT, BAG-DT and SVMs).

The novelty factor of the system also includes the unique hybrid approach to positioning. In order to benefit from both traditional positioning algorithms and Machine Learning, the authors propose a two-phase system. Machine Learning is initially used to narrow down specific areas where the tag might be, and then a separate algorithm is used for precision positioning.

For each set of data, two types of estimation methods have been used: Gradient Descent (noted with “X est” and “Y est”) and Least Squares (noted with “Xls_x” and “Xls_y).

For each set of data, the application calculates the difference in the timestamps between the master anchor and all other anchors and generates an input file for the predictor. The predictor generates an output file that is used by the application to retrieve the estimated sector/tile for each blink ID. After this file is parsed by the application, it will generate all possible combinations of three differences of timestamps and will apply both algorithms (Gradient Descent and Least Squares) for each possible combination. The result of each of these algorithms is then checked to fall within the correct area by comparing the resultant coordinates with the coordinates of the sector/tile generated by the predictor. If the output coordinates are within this sector/tile, these will be assumed to be valid, and an average of five values will be calculated as one estimation of the position. Both algorithms minimize the timestamp difference between three anchors, with the distinction that the Gradient Descent algorithm does this in an iterative way and Least Squares does this in a matrix form.

## 3. Results

The validation campaign consisted of two sets of tests, with one set of tests being conducted indoors and the other set of tests being conducted outdoors. In both situations, the same test equipment was used, composed of 10 anchors and 1 tag, which was placed in different positions.

### 3.1. Indoor Tests

The placement of the indoor anchors is shown in the screen capture in Figure 8, taken from the PC application developed for the test. The area is divided into 1 m × 1 m square tiles.

The UWB tag was placed in several subsequent positions, and the positioning result was plotted against the real position. The graphs show the coordinates in real distance values relative to the origin point (0,0). For each position, Anchor A1 was considered master. The results of the indoors test are summarized in Figure 9 where the Gradient Descent method is noted with “Est” and the Least Squares method is noted with Xls.

The variance of the error with the position of the tag is plotted in Figure 10 and Figure 11 for the X and Y axes, respectively.

The positioning error is independent of the tag position and remains, on average, under 0.5 m.

### 3.2. Outdoor Tests

The outdoor test was conducted in an area of 40 m × 10 m, which was divided into four sectors with a dimension of 10 m^2^, as seen in Figure 12. For each sector, four anchors were placed in the corners and were powered using 5V adapters. The master anchor was placed in position (x = 0, y = 0). The difference between the true position and the calculated position (determined by the AI from the tag messages TDoA information) was recorded for AI training and for performance assessment. Multiple tests were performed in Sector 1, for calibration purposes.

The UWB tag was placed in several subsequent positions in each sector corner (superimposed with the anchor positions) and in each sector center. The positioning result was plotted against the real position. The graphs show the coordinates in real distance values relative to the origin point (0,0). For each position, Anchor A1 was considered master.

#### 3.2.1. Using a Large 40 m × 10 m ML Predictor

One strategy for outdoor testing was to use a single, large ML predictor for the entire area (in this case, 40 m × 10 m). The results are plotted in Figure 13, where the Gradient Descent method is noted with “Est” and the Least Squares method is noted with Xls.

The variance of the error with the position of the tag is plotted in Figure 14 and Figure 15 for the X and Y axes, respectively.

It is apparent that the error increases as the tag is moved further away from the Master Anchor A1 on the X axis.

#### 3.2.2. Using a Small 10 m × 10 m ML Predictor

A second strategy was to use a smaller predictor, of 10 m × 10 m. In this second approach, the measurements were linearly translated from each position to the first sector, located at (X = 0; Y = 0), where the predictor and post processing were applied. After determining the position, the results were translated back into the original sector. These results are plotted in Figure 16, where the Gradient Descent method is noted with “Est” and the Least Squares method is noted with Xls.

The variance of the error with the position of the tag is plotted in Figure 17 and Figure 18 for the X and Y axes, respectively.

The results show less variance as the tag is moved away from the Master Anchor A1, compared to using the larger 40 m × 10 m predictor strategy.

## 4. Discussion

This study presents a UWB-based system, which has been developed for indoor and outdoor use. The system is composed of UWB tags and anchors and a software application. The underlying technology selected for providing raw distance measurements was UWB, making use of UWB tags and anchors manufactured by CDS.

The data processing chain includes AI/ML inference for rapid positioning in a 1 m × 1 m tile, coupled with optional post-processing based on classical algorithms, of which Gradient Descent and Least Squares were evaluated.Two types of functional tests were conducted, indoors and outdoors, and the results were presented in a synthetic form.

For indoors testing, a 12 m × 18 m area was used. The location was a live setup with IT equipment, furniture and active personnel, and it was heavily polluted with electromagnetic interference. One single ML predictor was used. The predictor was trained using artificial data generated by a TEE engine developed by CDS.

The indoor results indicate an accuracy better than 50 cm. These results are consistent and repeatable, and they are similar to results published in other recent research [12]. Consequently, the authors conclude that the system can be deployed and used commercially in real-life indoor setups.For outdoor testing, a 40 m × 10 m area was used. Two predictors were trained and used: one covering the entire area, and another covering only the first sector of 10 m × 10 m. The predictors were trained using artificial data generated by the TEE engine.When using the 10 m × 10 m predictor, the measurements were translated into the first sector, and after obtaining the AI/ML result and applying the post-processing, the results were translated back into the original sector.The outdoor results indicate an accuracy better than 1 m. These results are consistent and repeatable, and, consequently, the authors conclude that the system can be deployed and used commercially in real-life outdoor setups.The error analysis indicates that using a smaller, translatable 10 m × 10 m predictor yields better results than using a single, larger 40 m × 10 m predictor.The authors made available on GitHub [16] all raw measurements on which the graphs presented in this study are based, for easier reproduction of the results.

After reviewing the results of the validation tests, the authors conclude that one way to improve the system is to refine the post-processing algorithms applied after the AI/ML predictor has produced an initial result. In addition to Gradient Descent and Least Squares, other algorithms can be evaluated. Also, the filtering of raw UWB measurements can evolve into more sophisticated strategies, leading to increased result stability and accuracy.

It is known from the existing literature and from the authors’ own research that sub-10 cm accuracy can be obtained when performing Two-Way Ranging or Four-Way Ranging between two points. However, in the case of TDoA systems, real-life component imperfections, particularly crystals and oscillators, will multiply in positioning errors, which will end up being more than the stated 10 cm, unless high-end components and wired synchronization are used. In this study, the authors focused on a completely wireless TDoA-based system built with regular COTS components, which can handle hundreds of tags with minimal power consumption and high-frequency refresh rates. The results obtained are remarkable considering the constraints imposed; they are consistent, stable and perfectly usable in a series of commercial applications.

The analysis of the results and resulting conclusions are open for verification by the scientific community. The authors will make available upon request the measurement datasets with timestamped differences of arrival and ground truth positions for all the graphs shown in this study.

The tests can be replicated using the same off-the-shelf UWB tags and anchors and the same Machine Learning processing chain using AutoGluon on AWS.

The Machine Learning processing pipeline is run on the AWS environment, and its steps are described in Section 2.6 and Section 2.7, in sufficient detail as to allow replication.

The software used to automate the tests and present the results was developed by the authors specifically for this study using C# (version 12) and Java (version 21) languages and can be replicated with reasonable effort by a skilled practitioner of the software programming art.

## 5. Patents

Some methods described in this study, such as the methods for refining the positioning after obtaining the ML solution, are pending patent.

## Figures and Tables

**Figure 1 sensors-25-04620-f001:**
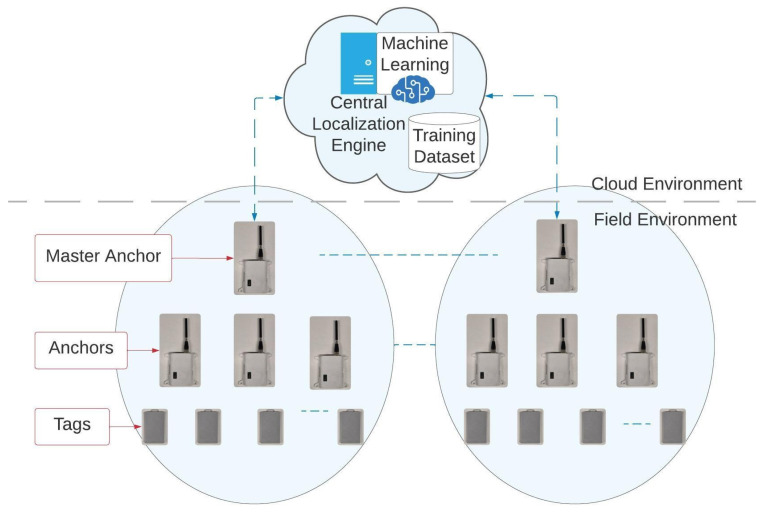
System architecture.

**Figure 2 sensors-25-04620-f002:**
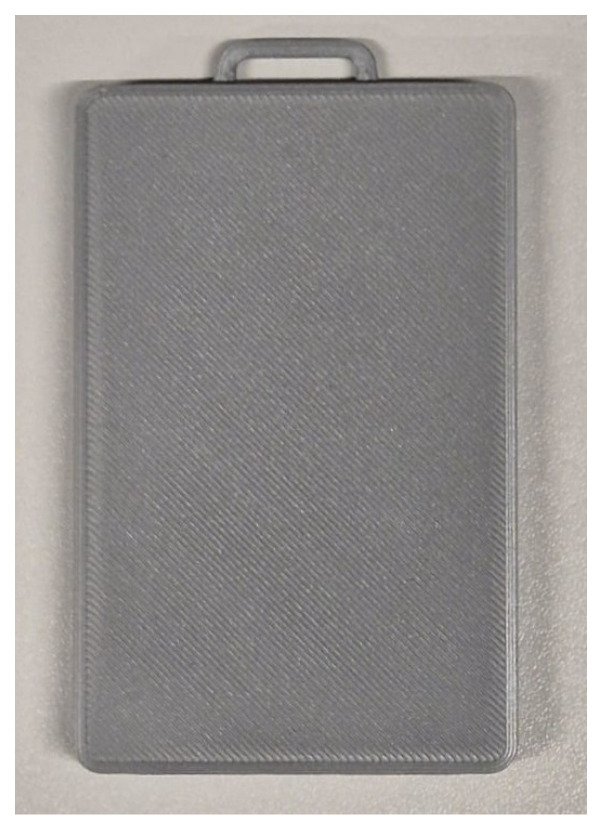
UWB tag.

**Figure 3 sensors-25-04620-f003:**
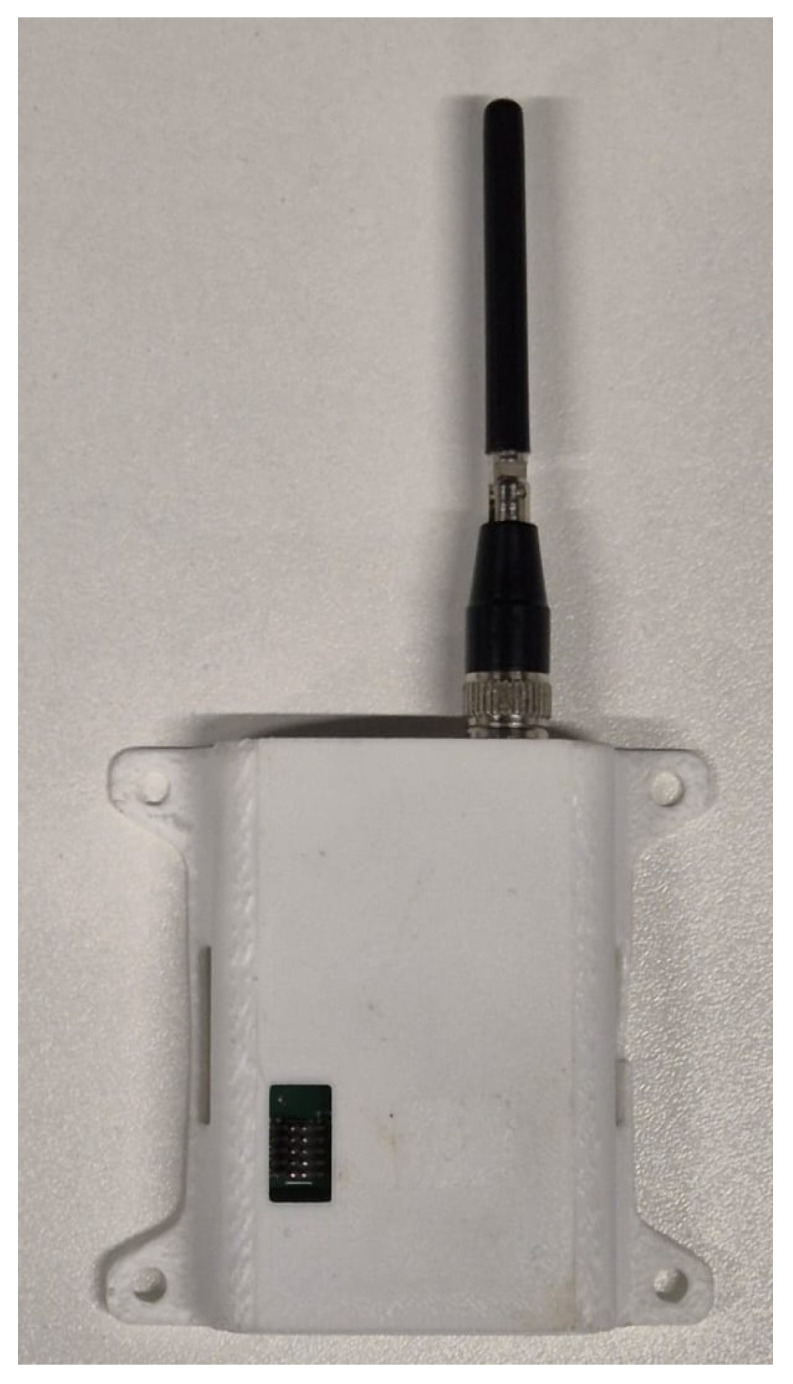
UWB anchor.

**Figure 4 sensors-25-04620-f004:**
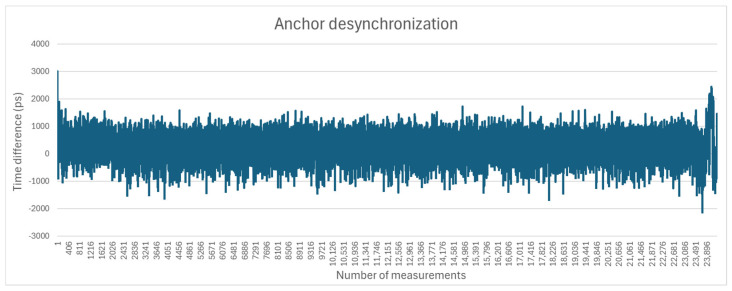
Anchor desynchronization.

**Figure 5 sensors-25-04620-f005:**
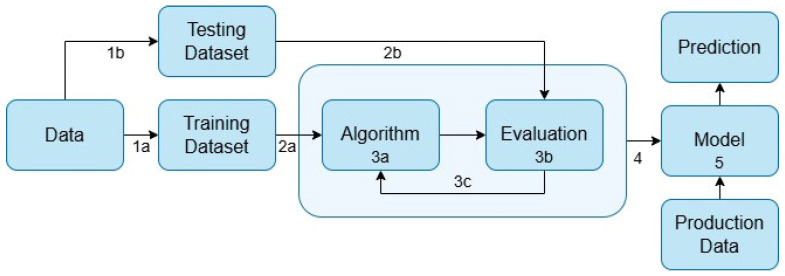
Machine Learning lifecycle.

**Figure 6 sensors-25-04620-f006:**
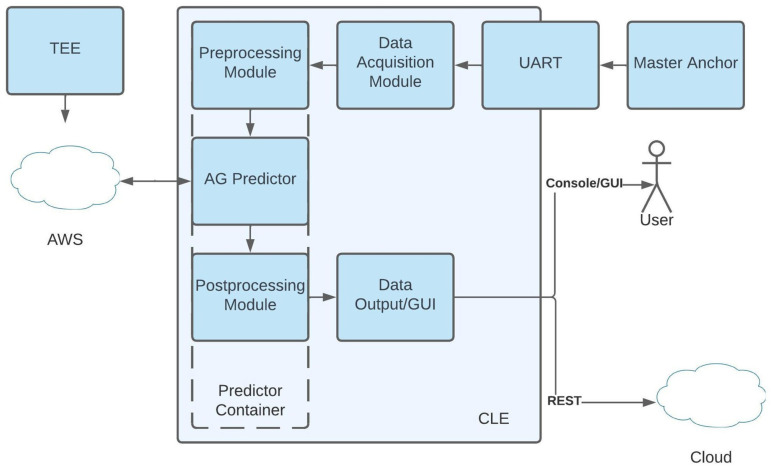
Software application architecture.

**Figure 7 sensors-25-04620-f007:**
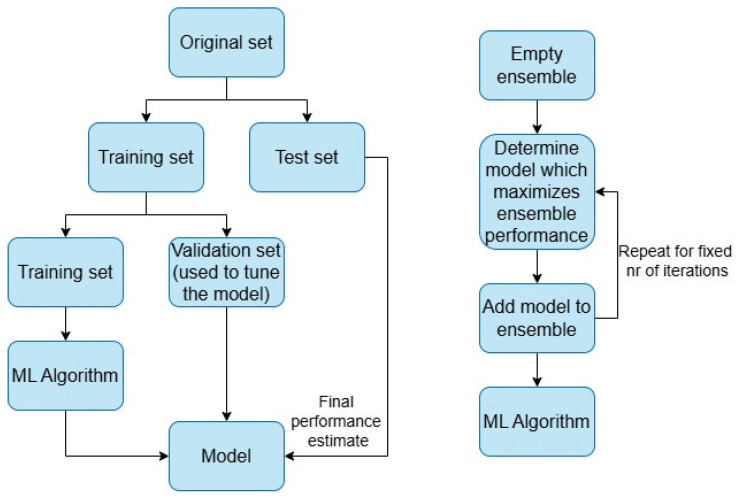
ML data processing.

**Figure 8 sensors-25-04620-f008:**
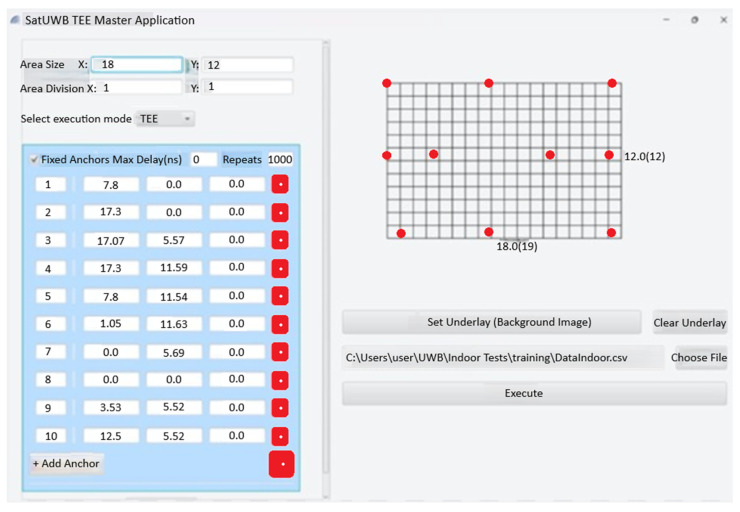
Indoor anchor placement.

**Figure 9 sensors-25-04620-f009:**
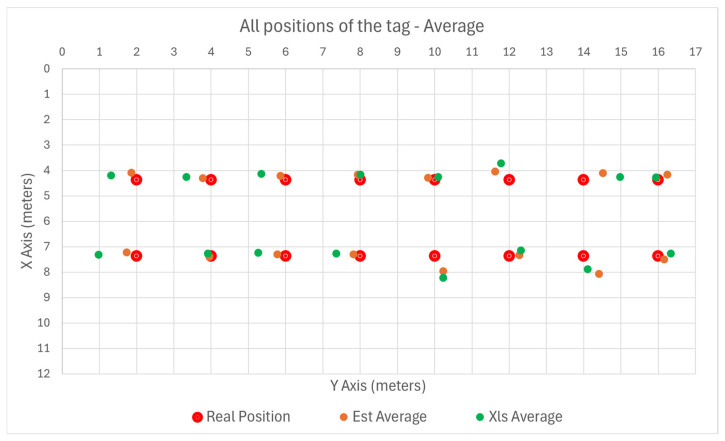
Indoor position plotting.

**Figure 10 sensors-25-04620-f010:**
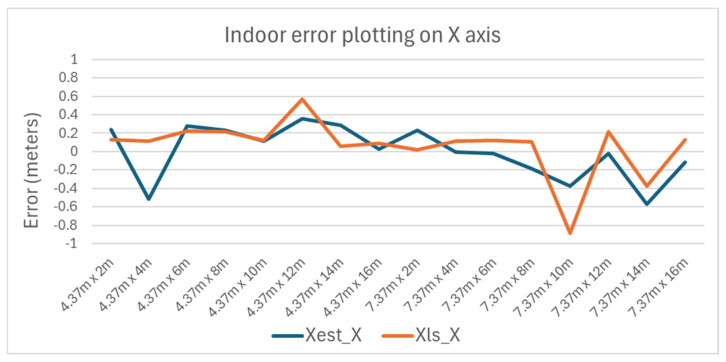
Indoor error plotting on X axis.

**Figure 11 sensors-25-04620-f011:**
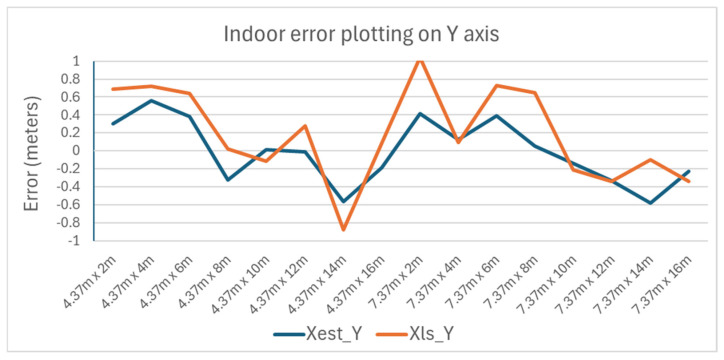
Indoor error plotting on Y axis.

**Figure 12 sensors-25-04620-f012:**
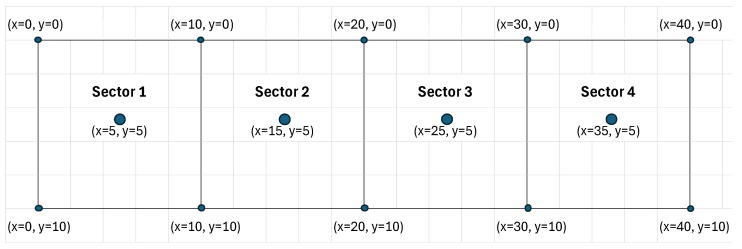
Outdoor anchor placement.

**Figure 13 sensors-25-04620-f013:**
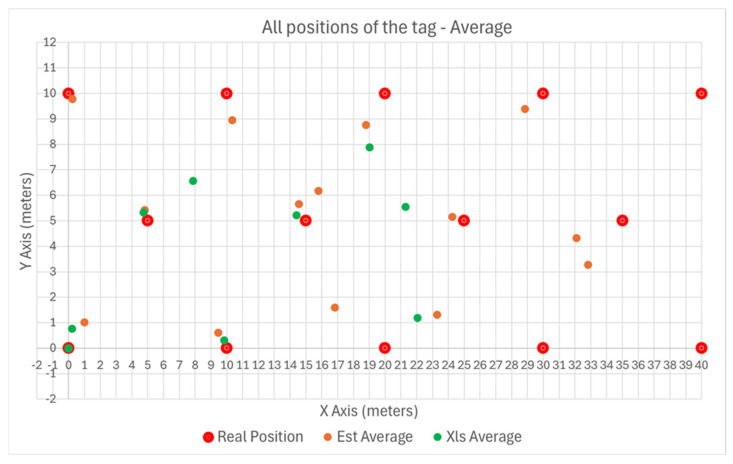
Outdoor position plotting, using a predictor of 40 m × 10 m.

**Figure 14 sensors-25-04620-f014:**
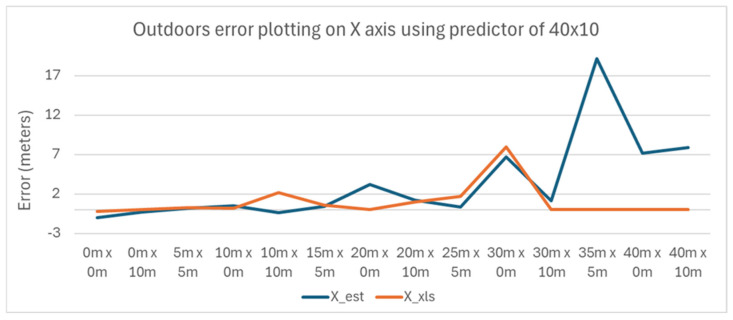
Outdoors error plotting on X axis using a predictor of 40 m × 10 m.

**Figure 15 sensors-25-04620-f015:**
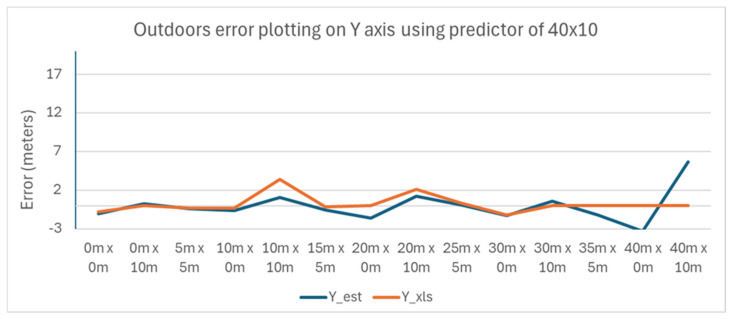
Outdoors error plotting on Y axis using a predictor of 40 m × 10 m.

**Figure 16 sensors-25-04620-f016:**
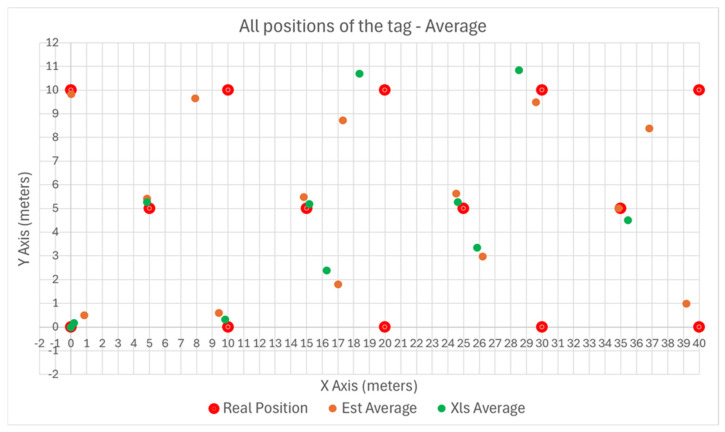
Outdoors position plotting, using a predictor of 10 m × 10 m.

**Figure 17 sensors-25-04620-f017:**
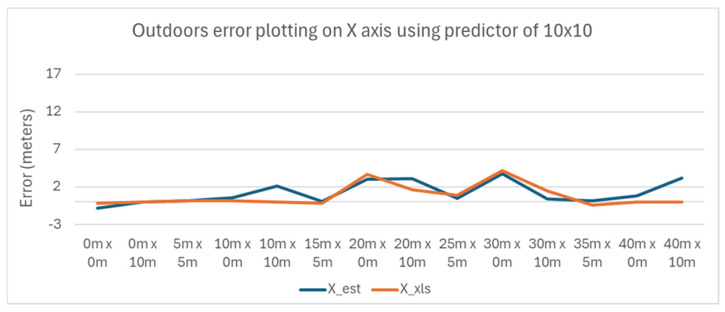
Outdoors error plotting on X axis using a predictor of 10 m × 10 m.

**Figure 18 sensors-25-04620-f018:**
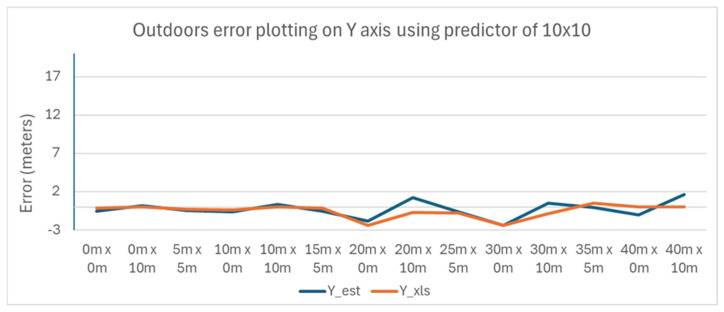
Outdoors error plotting on Y axis using a predictor of 10 m × 10 m.

**Table 1 sensors-25-04620-t001:** AutoGluon ML models.

Model	Description
AbstractModel	Abstract model implementation from which all AutoGluon models inherit
LGBModel	LightGBM model: https://lightgbm.readthedocs.io/en/latest/ (accessed on 1 November 2024)
CatBoostModel	CatBoost model: https://catboost.ai/ (accessed on 1 November 2024)
XGBoostModel	XGBoost model: https://xgboost.readthedocs.io/en/latest/ (accessed on 1 November 2024)
RFModel	Random Forest model (scikit-learn): https://scikit-learn.org/stable/modules/generated/sklearn.ensemble.RandomForestClassifier.html (accessed on 1 November 2024)
XTModel	Extra Trees model (scikit-learn): https://scikit-learn.org/stable/modules/generated/sklearn.ensemble.ExtraTreesClassifier.html#sklearn.ensemble.ExtraTreesClassifier (accessed on 1 November 2024)
KNNModel	KNearestNeighbors model (scikit-learn): https://scikit-learn.org/stable/modules/generated/sklearn.neighbors.KNeighborsClassifier.html (accessed on 1 November 2024)
LinearModel	Linear model (scikit-learn): https://scikit-learn.org/stable/modules/generated/sklearn.linear_model.LogisticRegression.html (accessed on 1 November 2024)
TabularNeuralNetModel	Class for neural network models that operate on tabular data.
NNFastAiTabularModel	Class for fastai v1 neural network models that operate on tabular data.

## Data Availability

The original contributions presented in this study are included in the article. Further inquiries can be directed to the corresponding author.

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
