# Peer review of "A Complete Solution for Ultra-Wideband Based Real-Time Positioning"

_sensors, 2025, doi:10.3390/s25154620_

Round 1

Reviewer 1 Report

Comments and Suggestions for Authors

In this paper, the authors proposed a complete, hardware and software, TDoA based real time positioning system. Some concerns are shown as following.

  1. In the abstract, the gap between the published work and the proposed work is not clearly explained. Therefore, the motivation of the proposed cannot convince readers.
  2. The introduction is too short to present the history of indoor localization because the corresponding algorithms and techniques have been developed several decades.
  3. The second section "Materials and Methods" only list the related techniques without any analysis and insights. What are the main bottleneck or disadvantages of the published solutions?
  4. What is the coverage of the proposed system architecture?
  5. Section 2.5 is too brief to clearly explain the ML localization. What is the main learning algorithm and what is the computational complexity?
  6. Section 2.6 includes details of software. What is the main contribution of the software implementation?
  7. Several figures in Section 2 are not clearly.
  8. The indoor anchor placement should include more detail such as the coverage size and distance among these anchors.
  9. How to evaluate the accuracy of the proposed localization scheme?

Author Response

Comment 1: In the abstract, the gap between the published work and the proposed work is not clearly explained. Therefore, the motivation of the proposed cannot convince readers.

Response 1: We agree with the suggestion. The abstract has been improved to present more clearly the improvement over the existing UWB RTLS systems, i.e., the combination of Machine Learning and classical positioning algorithmics.

Comment 2: The introduction is too short to present the history of indoor localization because the corresponding algorithms and techniques have been developed several decades.

Response 2: We understand the comment; however, the aim of the paper is not to present an exhaustive history of indoor localization, which indeed spans over decades. Instead, in this paper the authors are focusing specifically on UWB based systems. In section “2.1 Related Work” we refer to comprehensive literature specific to UWB systems ([5], [6], [7], [8], [9]) which should provide the reader with the background knowledge necessary to understand the domain. We are also identifying literature describing the argumentation for using UWB in the first place ([1], [2], [3], [4]). For the reader’s clarification, the introduction has been complemented with a clearer description of the solution differentiator.

Comment 3: The second section “Materials and Methods” only list the related techniques without any analysis and insights. What are the main bottleneck or disadvantages of the published solutions?

Response 3: We agree with the suggestion. The section “Materials and Methods” has been updated to include some of the main challenges of state of the art RTLS, along with the approach taken in this paper to address one in particular, i.e. the computational power required to run the positioning algorithms at high frequency rates for large size RTLS.

Comment 4: What is the coverage of the proposed system architecture?

Response 4: The question is pertinent, and the response adds to the reader’s benefit. The maximum distance between anchors varies from roughly 100m (line of sight, outdoor scenario) down to 20-30m (multiple obstacles and reflecting areas, indoor scenario). From the perspective of the computational power, which is the focus of this paper, the coverage area is divided into equal size rectangles, and their number has no impact on the resources needed to run the positioning algorithms and no impact on the Machine Learning pre-processing phase, except for the AI model training, performed at the beginning of the deployment. From the perspective of clock synchronization, the desynchronization between distant, multiple hop, anchors and the master anchor will introduce positioning errors which at some point will go over the targeted value. However, this is not the focus of the current paper, and the authors intend to research these limitations along with optimization techniques during future work. This clarification has been included in the new document revision under paragraph “2.8 Area coverage”.

Comment 5: Section 2.5 is too brief to clearly explain the ML localization. What is the main learning algorithm and what is the computational complexity?

Response 5: We understand the comment; however, the relevant information was included in section “2.6 Software Application”. To make things easier to understand, in the revised document we have moved some of the content from 2.6 to 2.5. It should now be more apparent to the reader that the authors have used AutoGluon on the AWS platform. The actual models used are listed in Table 1 of section 2.6.

Comment 6: Section 2.6 includes details of software. What is the main contribution of the software implementation?

Response 6: The software implementation has the purpose of a) acquiring the data from the master anchor according to the manufacturer specific API and b) automating the data processing and results presentation for purposes of performance evaluation. Both functions are implemented with common software development tools using either the Java or C# languages. While useful for experimentation, the software implementation does not contribute to the novelty of the solution and is therefore not described in detail. By contrast, the use of Machine Learning, specifically of the AutoGluon library, is considered central to this paper and is therefore described in detail.

Comment 7: Several figures in Section 2 are not clearly.

Response 7: We agree with the suggestion. The following figures have been updated in the revised document: 1,9,10,11,12,13,14,15,16,17.

Comment 8: The indoor anchor placement should include more detail such as the coverage size and distance among these anchors.

Response 8: This comment has been addressed in the answer to Comment 4 above, and clarifications have been added in the new document revision under paragraph “2.8 Area coverage”

Comment 9: How to evaluate the accuracy of the proposed localization scheme?

Response 9: In “Section 3 – Results”, the authors present the variance of the error with the tag position for each investigated scenario. These graphs are the synthesized result of multiple observation datasets, which the authors will provide upon request.

Reviewer 2 Report

Comments and Suggestions for Authors

The manuscript presents a real-time localization system based on Ultra-Wideband technology. The system is designed for both indoor and outdoor applications and includes hardware components, a synchronization mechanism, and a hybrid positioning approach combining traditional TDoA-based algorithms with machine learning techniques. The ML models are trained using data generated through a proprietary simulation tool. Validation tests are conducted in controlled environments, with positioning errors reported for both indoor (approx. 50 cm) and outdoor (up to 1 meter) scenarios. My comments are:

1) The introduction lacks a comprehensive and critical discussion of the state of the art. While a few references are included, the manuscript does not effectively position the proposed system with respect to existing solutions. A more rigorous review of current technologies, especially recent developments in UWB-based localization and sensor fusion, is needed.

2) Although the title claims a “complete solution,” the manuscript does not provide enough technical details to ensure reproducibility. Key aspects such as algorithmic steps, data preprocessing pipelines, and model configurations are either missing or only superficially described. The machine learning components are presented with generalities, without offering access to the datasets, code, or parameter settings used.

3) The figures are generally blurred and lack sufficient resolution. This severely limits their ability to support the technical narrative and interpret the experimental results. High-quality plots with clear legends and axis labels are essential.

4) The reported positioning errors—up to 80 cm in some cases are significantly worse than what is typically achieved by modern UWB systems, which routinely offer sub 10 cm accuracy. When combined with inertial sensor fusion, performance can be further improved. The manuscript does not acknowledge this gap nor propose strategies to close it.

5) The reference list is short and does not reflect the breadth of relevant research in the area. Several seminal and recent contributions to UWB-based RTLS, hybrid localization, and learning based methods are not cited. This further weakens the scientific framing of the work.

Comments on the Quality of English Language

The manuscript is written in poor English, with numerous grammatical errors, typos, and awkward phrasing. The text is often repetitive and occasionally confusing, which detracts from the clarity and professionalism expected in a scientific journal.

Author Response

Comment 1: The introduction lacks a comprehensive and critical discussion of the state of the art. While a few references are included, the manuscript does not effectively position the proposed system with respect to existing solutions. A more rigorous review of current technologies, especially recent developments in UWB-based localization and sensor fusion, is needed.

Response 1: We understand the comment; however, in section “2.1 Related Work” the authors do refer to comprehensive literature specific to UWB systems ([5], [6], [7], [8], [9]) which should provide the reader with the background knowledge necessary to understand the domain and the state of the art. The authors are also identifying literature describing the argumentation for using UWB in the first place ([1], [2], [3], [4]). In this paper, the authors focus on UWB based RTLS; more precisely, the focus is on the implementation and validation of a specific improvement method (the use of Machine Learning as a pre-processing step), rather than presenting the reader with a comprehensive survey of existing solutions, which, although can constitute a perfectly valid topic, would be the subject of a different paper. For the reader’s clarification, the introduction section has been complemented with a clearer description of the solution differentiator.

Comment 2: Although the title claims a “complete solution,” the manuscript does not provide enough technical details to ensure reproducibility. Key aspects such as algorithmic steps, data preprocessing pipelines, and model configurations are either missing or only superficially described. The machine learning components are presented with generalities, without offering access to the datasets, code, or parameter settings used.

Response 2: The analysis of the results and resulting conclusions are open for verification by the scientific community. The authors will make available upon request the measurement datasets with timestamped differences of arrival and ground truth positions for all the graphs shown in the paper. The tests can be replicated using the same off the shelf UWB tags and anchors, and the same Machine Learning processing chain using AutoGluon on AWS. The Machine Learning processing pipeline is running on the AWS environment and its steps are described in paragraphs “2.6 Software application”, and “2.7 Model training”, in sufficient detail as to allow replication. The software used to automate the tests and present the results has been developed by the authors specifically for this study using C# and Java languages and can be replicated with reasonable effort by a skilled practitioner of the software programming art.

Comment 3: The figures are generally blurred and lack sufficient resolution. This severely limits their ability to support the technical narrative and interpret the experimental results. High-quality plots with clear legends and axis labels are essential.

Response 3: We agree with the suggestion. The following figures have been updated in the revised document: 1,9,10,11,12,13,14,15,16,17.

Comment 4: The reported positioning errors—up to 80 cm in some cases are significantly worse than what is typically achieved by modern UWB systems, which routinely offer sub 10 cm accuracy. When combined with inertial sensor fusion, performance can be further improved. The manuscript does not acknowledge this gap nor propose strategies to close it.

Response 4: We agree with the comment and confirm through our own research that sub 10 cm accuracy can be obtained when performing Two Way Ranging or Four Way Ranging techniques between two points. However, in the case of TDoA systems, real life component imperfections, particularly crystals and oscillators, will multiply in positioning errors which will end up being more than the stated 10cm, unless high end components and wired synchronization are used. In this paper we have focused on a completely wireless TDoA based system built with regular COTS components which can handle hundreds of tags with minimal power consumption and high frequency refresh rates. The results obtained are remarkable considering the constraints imposed, they are consistent, stable and perfectly usable in a series of commercial applications. For reader’s clarification this information has been included in section “4. Discussion” of the revised document.

Comment 5: The reference list is short and does not reflect the breadth of relevant research in the area. Several seminal and recent contributions to UWB-based RTLS, hybrid localization, and learning based methods are not cited. This further weakens the scientific framing of the work.

Response 5: We understand the comment. However, in section “2.1 Related Work” we refer to comprehensive literature specific to UWB systems ([5], [6], [7], [8], [9]) which should provide the reader with the background knowledge necessary to understand the domain and the state of the art. We are also identifying literature describing the argumentation for using UWB in the first place ([1], [2], [3], [4]). We remain open to improving the reference list based on concrete examples.

Comment 6: The manuscript is written in poor English, with numerous grammatical errors, typos, and awkward phrasing. The text is often repetitive and occasionally confusing, which detracts from the clarity and professionalism expected in a scientific journal.

Response 6: We understand the comment and have re-checked the wording and grammar, as well as for repetitions. We remain open to improving on specific paragraphs if the English imperfections are clearly identified.

Reviewer 3 Report

Comments and Suggestions for Authors

Point 1: The abstract is not focused on the problem to be described, and does not clearly state what scientific problem this paper solves. Authors need to streamline the content of abstract.

Point 2: The manuscript mainly uses existing techniques and algorithm, and has engineering application value, but its own innovation is not very clear.

Point 3: The background information on the clock compensation mechanism mentioned in this article is insufficient. Compared with other existing synchronization technologies, what is the accuracy performance of the time synchronization method in the manuscript?

Point 4: The clarity of most of the figures is not high enough to be seen clearly. It is suggested to improve the quality of figures, which will help readers understand.

 Point 5: The paper has a small written error on line 223: "Error! Reference source not found.”

Author Response

Comment 1: The abstract is not focused on the problem to be described, and does not clearly state what scientific problem this paper solves. Authors need to streamline the content of abstract.

Response 1: We agree with the suggestion, and the abstract has been improved to more clearly present the differentiation against the existing UWB RTLS systems, i.e., the combination of Machine Learning and classical positioning algorithmics.

Comment 2: The manuscript mainly uses existing techniques and algorithm, and has engineering application value, but its own innovation is not very clear.

Response 2: We agree with the suggestion. To clarify, the use of Machine Learning as a preprocessing step is an innovative technique which, after the model is trained, reduces the computational power required to reach an algorithmic solution. This clarification has been introduced in the revised document in the “Abstract”, “Introduction” and “Discussion” sections.

Comment 3: The background information on the clock compensation mechanism mentioned in this article is insufficient. Compared with other existing synchronization technologies, what is the accuracy performance of the time synchronization method in the manuscript?

Response 3: The observation is valid, however the accuracy of the clock synchronization method, although recognized as of high importance to the overall positioning accuracy, is not the subject of the current paper. Clock synchronization in multi-hop wireless RTLS, along with optimization techniques, is the topic of future work.

Comment 4: The clarity of most of the figures is not high enough to be seen clearly. It is suggested to improve the quality of figures, which will help readers understand.

Response 4: We agree with the suggestion. The following figures have been updated in the revised document: 1,9,10,11,12,13,14,15,16,17.

Comment 5: The paper has a small written error on line 223: "Error! Reference source not found.”

Response 5: The observation is correct; the text has been updated in the revised document.

Round 2

Reviewer 1 Report

Comments and Suggestions for Authors

The authors revised the manuscript based on reviewers' comments.

Author Response

Comment 1: The authors revised the manuscript based on reviewers' comments.

Response 1: Thank you, we have tried to address all the comments. We can perform further modifications to the text in case specific paragraphs are identified.